# CD229 CAR T cells eliminate multiple myeloma and tumor propagating cells without fratricide

Sabarinath V. Radhakrishnan[1,7], Tim Luetkens [1,7]*, Sandra D. Scherer[2], Patricia Davis[3], Erica R. Vander Mause[1], Michael L. Olson [1], Sara Yousef[1], Jens Panse[4], Yasmina Abdiche[5], K. David Li[3], Rodney R. Miles[3], William Matsui[6], Alana L. Welm[2] & Djordje Atanackovic[1]

Multiple myeloma (MM) is a plasma cell malignancy and most patients eventually succumb to the disease. Chimeric antigen receptor (CAR) T cells targeting B-Cell Maturation Antigen (BCMA) on MM cells have shown high-response rates, but limited durability. CD229/LY9 is a cell surface receptor present on B and T lymphocytes that is universally and strongly expressed on MM plasma cells. Here, we develop CD229 CAR T cells that are highly active in vitro and in vivo against MM plasma cells, memory B cells, and MM-propagating cells. We do not observe fratricide during CD229 CAR T cell production, as CD229 is down-regulated in T cells during activation. In addition, while CD229 CAR T cells target normal CD229$^{high}$ T cells, they spare functional CD229$^{neg/low}$ T cells. These findings indicate that CD229 CAR T cells may be an effective treatment for patients with MM.

[1] Multiple Myeloma Program & Cancer Immunotherapy, Division of Hematology and Hematologic Malignancies, Huntsman Cancer Institute, University of Utah, Salt Lake City, UT 84112, USA. [2] Department of Oncological Sciences, Huntsman Cancer Institute, University of Utah, Salt Lake City, UT 84112, USA. [3] Department of Pathology, University of Utah and ARUP Laboratories, Salt Lake City, UT, USA. [4] Department of Oncology, Hematology, Hemostaseology, and Stem Cell Transplantation, University Hospital RWTH Aachen, Aachen, Germany. [5] Carterra Inc., Salt Lake City, UT, USA. [6] Department of Oncology, The University of Texas at Austin, Austin, TX, USA. [7] These authors contributed equally: Sabarinath V. Radhakrishnan, Tim Luetkens. *email: tim.luetkens@hci.utah.edu

Multiple myeloma (MM) is a clonal plasma cell (PC) malignancy that develops in the patients' bone marrow (BM) and ultimately causes bone lesions with fractures, kidney failure, BM failure, and immunoparesis with fatal infections. Approximately 30,000 new cases are diagnosed per year in the United States alone[1] and almost all patients will eventually succumb to the disease due to the development of chemotherapy resistance.

T cells engineered to express chimeric antigen receptors (CARs) can be used to effectively target tumor cells. CARs combine a binding domain against a surface antigen with signaling domains inducing T cell activation. The adoptive transfer of CAR T cells targeting B cell maturation antigen (BCMA) has resulted in high overall response rates in MM patients. However, modulation of BCMA expression after BCMA CAR T cell therapy has been observed and median progression-free survival is <1 year[2,3]. In addition, 33/85 patients screened for a clinical BCMA CAR T-cell trial did not express BCMA in their bone marrow[4]. Currently, there is no cellular immunotherapy available for the substantial number of patients who are not eligible for anti-BCMA CAR T cells in the first place or patients with relapses after BCMA CAR T cell treatment.

It has previously been shown that the SLAM receptor CD229/LY9 is a potential target for CAR T cell therapy in MM due to its strong and homogenous expression on the bulk of tumor cells, as well as chemotherapy-resistant myeloma progenitors, its absence from most normal cells, and dependence of MM cells on CD229 for their survival[5–8]. We now develop CD229-specific CAR T cells showing strong and persistent activity against MM in vitro and in vivo.

## Results

### Expression of CD229 in MM plasma cells and B lineage cells.

Analyzing bone marrow samples from 20 MM patients using flow cytometry, we found that, in agreement with previous observations[5,7–9], CD229 shows equally strong expression on the surface of MM cells from all patients with newly diagnosed and relapsed/refractory disease (Fig. 1a). It has been suggested that clonotypic MM B cells may be present in the pre-plasma cell memory B cell compartment[10–12] and targeting of CD19+ B cells in MM patients using CAR T cells has shown some clinical activity[13,14]. We found that CD229, but not BCMA expression, can already be detected in transitional and memory B cells (Fig. 1b). This finding suggests that targeting CD229 may lead to the eradication not only of terminally differentiated MM plasma

cells but also clonotypic MM-propagating cells potentially present in the memory B-cell compartment.

### Generation of a fully human CD229-specific antibody.

We next generated the first fully human antibody against CD229 using antibody phage display. In an initial monoclonal screening of the selected binders using a time-resolved fluorescence-based binding assay (Supplementary Fig. 1A), we identified 23 antibodies recognizing CD229 and lacking measurable affinity to any other SLAM family receptors (Supplementary Fig. 1B). We then stained 293T cells expressing second generation CARs (Supplementary Fig. 1C) based on these antibodies with recombinant CD229 and observed high-surface expression and antigen binding by 15/23 constructs (Supplementary Fig. 1D). For subsequent assays, we selected clone 2D3 (Supplementary Fig. 1E), which showed strong CD229 binding in monoclonal antibody and CAR format (Supplementary Fig. 1F) and no off-target binding to a set of 5,300 human surface proteins (Fig. 2a). In addition, we observed binding of 2D3 to K562 cells transduced with a CD229 expression construct (K562-CD229) but not to CD229⁻ parental K562 cells (Fig. 2b). Replacing the four Ig-like domains in the extracellular region of human CD229 with their mouse homologs, we mapped the binding region of 2D3 to the membrane-proximal variable domain of CD229 (Fig. 2c).

Determining the selectivity of 2D3 for MM cells compared to normal hematopoietic cells, we first analyzed binding of 2D3 to purified MM plasma cells using flow cytometry and observed strong and uniform staining of all malignant cells (Fig. 2d). Comparing binding of 2D3 to MM cells in bone marrow samples from patients with newly diagnosed disease to patients with relapsed/refractory disease, we again observed universal staining and no significant difference between the groups (Fig. 2e). We next determined binding of 2D3 to other hematopoietic cells and did not observe any binding to CD34+ hematopoietic progenitors, neutrophils, or monocytes (Fig. 2f). Analyzing normal lymphocyte subpopulations, 2D3 did not show any binding to NK cells but weakly stained T and B cells (Fig. 2g).

### CD229 CAR T cells show minimal fratricide.

We next generated primary human CD229 CAR T cells based on 2D3 (outline of production process in Supplementary Fig. 2B) and demonstrated efficient surface expression of the CD229 CAR construct using flow cytometry (Supplementary Fig. 2C). As we had observed binding by 2D3 to T cells, we asked the question whether 2D3-based CAR T cells would exhibit fratricide as previously observed

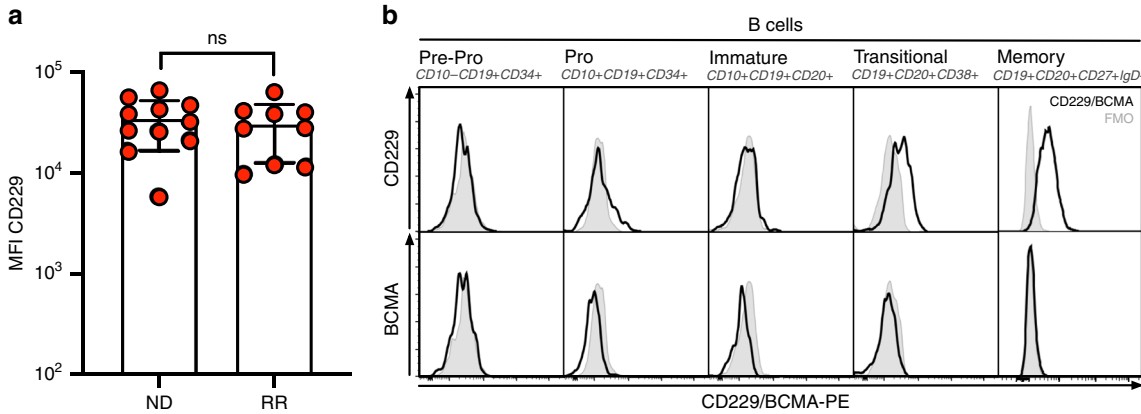

**Fig. 1 CD229 expression on MM plasma cells and B lineage cells. a** Expression of CD229 on CD38+CD138+ plasma cells in the bone marrow from 20 patients with MM as determined by flow cytometry after staining with HLy9.1.25. Data represent mean ± SD. ND newly diagnosed, RR relapsed/refractory. Statistical significance was determined by two-sided Student's t-test. **b** Expression of CD229 and BCMA on B cell populations in the bone marrow of a patient with MM as determined by flow cytometry. FMO fluorescence minus one. Source data are provided as a Source Data file.

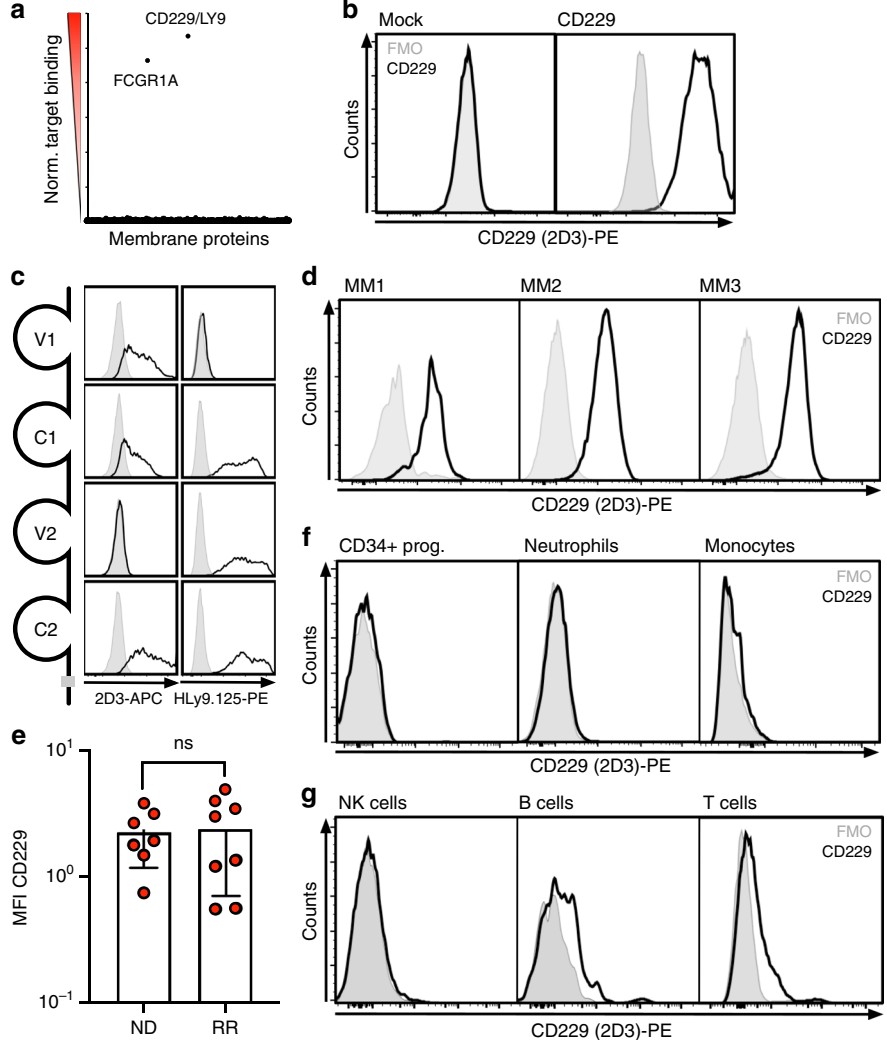

**Fig. 2 The fully human anti-CD229 antibody 2D3 binds to MM cells and B and T cells. a** Binding of the fully human IgG₁ antibody 2D3 to 293 cells individually expressing 5300 full-length, human membrane proteins as determined by flow cytometry. Fc gamma receptor 1A (FCGR1A) serves as a positive control. **b** K562 cells were stably transduced with a human CD229 expression construct and stained with 2D3. Binding was determined by flow cytometry. **c** *Left*: The extracellular domain of CD229 consists of two variable and two constant Ig-like domains. In a human CD229 expression construct, each of these domains was replaced with the corresponding mouse homolog as 2D3 did not show any reactivity to mouse CD229 (Supplementary Fig. 2A). *Right*: Binding by 2D3 and the commercially available CD229-specific antibody HLy9.1.25[33] to the four chimeric proteins expressed on 293T cells was determined by flow cytometry. **d** MM cells were isolated from primary bone marrow samples from three patients with MM using a commercially available MM cell purification kit (Stemcell Technologies). Purified MM cells were stained with 2D3 and binding was determined by flow cytometry. **e** Bone marrow samples from seven patients with newly diagnosed (ND) and eight patients with relapsed/refractory (RR) MM were stained with 2D3 and analyzed by flow cytometry. Data represent mean ± standard deviation and statistical significance was determined by two-sided Student's *t*-test. **f** CD34⁺ hematopoietic progenitors, neutrophils, monocytes, **g** natural killer (NK) cells, B cells, and T cells were isolated from peripheral blood from a healthy donor using commercially available purification kits (Stemcell Technologies). Cells were stained with 2D3 and analyzed by flow cytometry. Source data are provided as a Source Data file.

in CAR T cells targeting surface antigens expressed on healthy T cells[15,16]. However, we did not observe a significant reduction in the expansion of CD229 CAR T cells compared to control T cells expressing either a CAR without a binding domain (ΔscFv) or CD19 CAR T cells[17,18] (Fig. 3a). Comparing phenotypes of CAR T cells at the end of manufacturing by flow cytometry, we also observed no significant differences between CD229 CAR T cells and CD19 CAR T cells with most of the T cells showing a central memory phenotype (Supplementary Fig. 2D).

Analyzing CD229 CAR T-cell function, we found that CD229 CAR T cells secreted pro-inflammatory cytokines (Supplementary Fig. 2E)[19–21] and proliferated in the presence of CD229⁺ target cells (Supplementary Fig. 2F). We next measured the cytotoxic

activity of CD229 CAR T cells against K562 and K562-CD229 cells and observed specific killing only of the CD229-expressing cells (Fig. 3b). Further analyzing the possibility of fratricide, which means killing of CAR T cells by other CAR T cells, we observed no cytotoxic activity of CD229 CAR T cells against activated T cells (T_act) that had undergone the same CAR T cell production process except for the addition of lentiviral particles (Fig. 3c). Comparing CD229 expression in normal T cells and T_act cells, we found that T_act cells had downregulated CD229 protein (Fig. 3d) and mRNA (Fig. 3e) expression, resulting in the loss of 2D3 binding (Fig. 3f). While we did not observe substantial levels of fratricide or targeting of NK cells (Supplementary Fig. 3A), we found that, without prior T-cell activation, CD229 CAR T cells

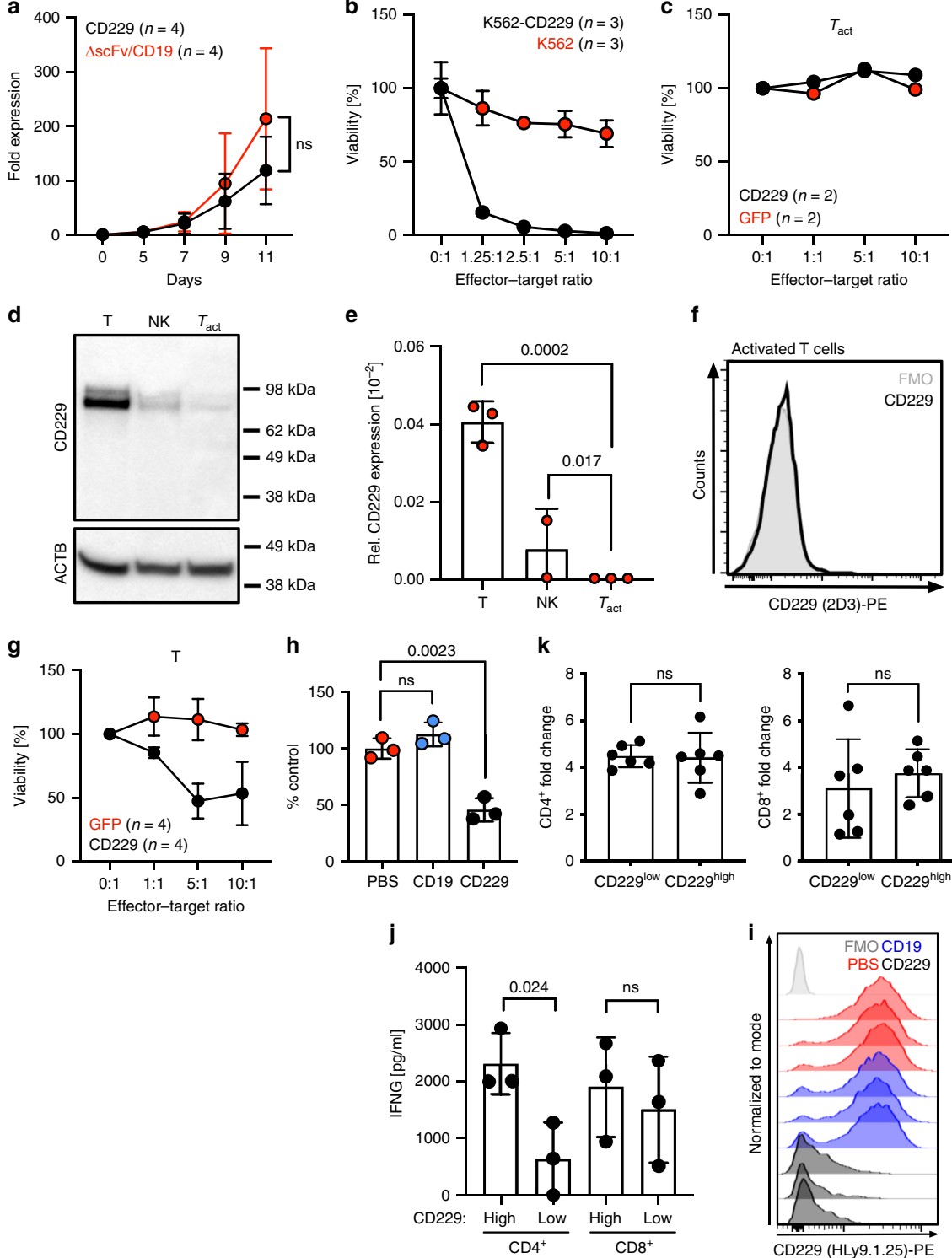

targeted normal T cells in vitro (Fig. 3g) and in an in vivo cytotoxicity assay using human peripheral blood mononuclear cells (PBMCs) (Fig. 3h and Supplementary Fig. 3B). We observed that the remaining T cells in our in vivo experiment were CD229[neg/low] (Fig. 3i), and determined that emergence of this population was not the result of CD229 downregulation induced by CD229 CAR T cells but of the sparing of CD229[neg/low] T cells (Supplementary Fig. 3C). A similar observation has previously been made by Gogishvili et al.[22] who found that CAR T cells

targeting the SLAM receptor CS1 selected a functional CS1[neg] T-cell population.

Asking the question whether non-activated CD229[neg/low] T cells are functional, we next compared CD229[high] to CD229[neg/low] T cells. We observed reduced IFNG secretion by CD4+ (Fig. 3j) but not by CD8+ (Fig. 3j) CD229[neg/low] T cells. However, we did not observe significant differences in T-cell phenotypes (Supplementary Fig. 3D) and proliferation in response to stimulation with CD3/CD28 beads (Fig. 3k),

**Fig. 3 Production of CD229 CAR T cells and T-cell targeting. a** CAR T-cell expansion as determined by automated cell counting. Data represent mean ± SD from four healthy donors. Significance of differences between cell numbers on day 11 was determined by two-sided Student's t-test. **b** Killing of K562 cells transduced with a CD229 expression construct or parental CD229⁻ K562 cells by CD229 CAR T cells as determined by luciferase-based cytotoxicity assay. Data represent mean ± SD from three independent experiments. **c** Cytotoxic activity of CD229 CAR T cells against T cells that had undergone CAR T-cell manufacturing for 7 days except for addition of lentiviral particles ($T_{act}$) as determined by flow cytometry. Data represent mean from two independent experiments. **d** CD229 expression in whole cell lysates from T cells, NK cells, and $T_{act}$ cells, as determined by western blot. Blots show representative results from two independent experiments. **e** CD229 expression in T cells, NK cells, and $T_{act}$ cells as determined by quantitative RT-PCR. Data represent mean ± SD from three replicates. Statistical significance was determined by two-sided Student's t-test. **f** Binding of 2D3 to $T_{act}$ cells. Histogram shows representative result for one healthy donor from at least 3 independent experiments. **g** In vitro cytotoxicity of CD229 CAR T cells or GFP T cells against purified healthy T cells as determined by flow cytometry. Data represent mean ± SD of three independent experiments. **h** NSG mice were intravenously injected with $10^7$ healthy PBMCs. After 2 days, mice were injected with $1 \times 10^6$ CD19 or CD229 CAR T cells or PBS and sacrificed after 4 days. Data represent mean ± SD of healthy non-CAR T cell numbers in the peripheral blood from three animals per group. Significance was determined by two-sided Student's t-test. **i** CD229 expression on T cells from the same animals as determined by flow cytometry. **j** T cells from healthy donors were sorted for CD229 expression (Supplementary Fig. 3E) and stimulated with CD3/CD28 beads and 40 IU/ml IL-2 for 4 days. IFNG secretion by CD229$^{low}$ and CD229$^{high}$ T cells was determined by cytometric bead array (BD). Data represent mean ± SD from three healthy donors. Significance was determined by two-sided Student's t-test. **k** Expansion of CD229$^{low}$ and CD229$^{high}$ T cells as determined by automated cell counting. Data represent mean ± SD from six healthy donors. Significance was determined by two-sided Student's t-test. Source data are provided as a Source Data file.

indicating that CD229$^{neg/low}$ T cells are functional and exhibit key effector functions.

**CD229 CAR T cells eliminate MM including MM-propagating cells.** We next analyzed the cytotoxic activity of CD229 CAR T cells against MM cells. Using two MM cell lines, U-266 and RPMI-8226, expressing different levels of CD229 (Suppl. Fig. 4A), we found that CD229 CAR T cells but not ΔscFv CAR T cells efficiently killed both cell lines in vitro (Fig. 4a). Furthermore, using primary CD138$^+$ tumor cells from 3 patients with plasma cell leukemia, a highly aggressive form of MM, which all showed high expression of CD229 (Suppl. Fig. 4B), we found that CD229 CAR T cells exhibited high cytotoxic activity against these cells (Fig. 4b) and pro-inflammatory cytokine production (Fig. 4c). In order to determine the in vivo activity of CD229 CAR T cells, we performed two NOD.Cg-Prkdc$^{scid}$ Il2rg$^{tm1Wjl}$/SzJ (NSG) xenograft models. After engraftment of the tumor cells for 7 days, we injected a single dose of $3 \times 10^6$ CAR T cells and monitored animals using in vivo imaging (Supplementary Fig. 4C). We found that in mice engrafted with MM cell lines U-266 (Fig. 4d) and RPMI-8226 (Fig. 4e), treatment with CD229 CAR T cells resulted in the loss of luminescence signal in 5/5 and 4/5 animals, respectively, a significant reduction in MM-related serum light-chain levels (Supplementary Fig. 4D), and significantly prolonged survival.

CD19-specific CAR T cells have previously been shown to be effective in some patients with MM[13,14] although CD19 is typically absent from most MM plasma cells. It has been hypothesized that the therapeutic benefit of CD19 CAR T cells in patients with MM is the result of the targeting of clonotypic MM-propagating cells present in the B cell compartment[13]. As expected based on our binding data, we found that CD229 CAR T cells killed B lymphocytes in vitro (Fig. 4f). In our humanized mouse model, CD229 CAR T cells also efficiently targeted B cells in the animals' peripheral blood (Fig. 4g) and bone marrow (Fig. 4h). Specifically, CD229 CAR T cells spared naive B cells but killed memory B cells (Fig. 4i). In contrast, BCMA CAR T cells, which specifically bound to (Supplementary Fig. 4E) and killed MM cells (Supplementary Fig. 4F), did not target either B cell population (Fig. 4i). Based on these findings, we hypothesized that CD229 CAR T cells may be able to target not only MM plasma cells but also clonotypic MM-propagating cells present in the memory B-cell population. To explore this possibility, we performed colony formation assays to quantify MM-propagating cells in BM samples from seven patients with MM after ΔscFv, BCMA, or CD229 CAR T-cell treatment. While we found no

effect on normal hematopoietic progenitors (Supplementary Fig. 4G), we indeed observed a significant reduction in the number of MM-propagating cells in samples treated with CD229 CAR T cells compared to BCMA CAR T cells (Fig. 4j).

## Discussion

The standard of care for patients with MM includes treatments such as proteasome inhibitors, immunomodulatory drugs, autologous stem cell transplantation, and monoclonal antibodies, which have substantially prolonged patient survival[23]. However, to date, MM still remains an incurable disease. CAR T cells have emerged as an effective approach for a number of other hematologic malignancies and recent results from BCMA-targeting CAR T-cell clinical trials in MM indeed showed impressive overall response rates[2,3]. However, the durability of these responses was relatively limited and even patients with initial complete responses eventually relapsed. Relapses may be due to a number of factors including heterogenous and/or low pretreatment expression of BCMA by fully differentiated tumor cells[4], absence of BCMA from less differentiated MM cells, and loss of BCMA expression under the selective pressure of the CAR T cell-mediated immune response[2].

We have now developed CAR T cells against CD229, a surface antigen which shows universal and strong tumor expression in patients with MM[5–8]. CD229 CAR T cells efficiently eliminated not only terminally differentiated MM plasma cells, which are also targeted by BCMA CAR T cells, but also memory B cells, a potential reservoir for clonotypic MM cells[10–12] and MM-propagating cells. Combined with the essential function of CD229 in MM[7], we hypothesize that this broader targeting of MM cells may lead to more robust and durable clinical responses. A common mechanism of resistance to CAR T cell therapy in MM and other hematologic malignancies is the escape of tumor cells expressing lower levels of the target antigen[24]. Our data also indicate that CD229$^{low}$ lymphocytes and MM cells are not targeted as efficiently as CD229$^{high}$ cells by CD229 CAR T cells. We hypothesize that, although we and others have shown that CD229 is generally expressed at high levels on patient MM cells[5,7–9], antigen-negative relapses after CD229 CAR T-cell therapy are possible.

Like all SLAM receptors, CD229 also shows expression on normal lymphocytes but in contrast to CS1, the target of the FDA-approved monoclonal antibody elotuzumab and a clinical CAR T-cell trial[25], CD229 is absent from normal monocytes and dendritic cells[7,26]. Indeed, we found that CD229 CAR T cells showed cytotoxicity against CD229$^{high}$ T cells but spared

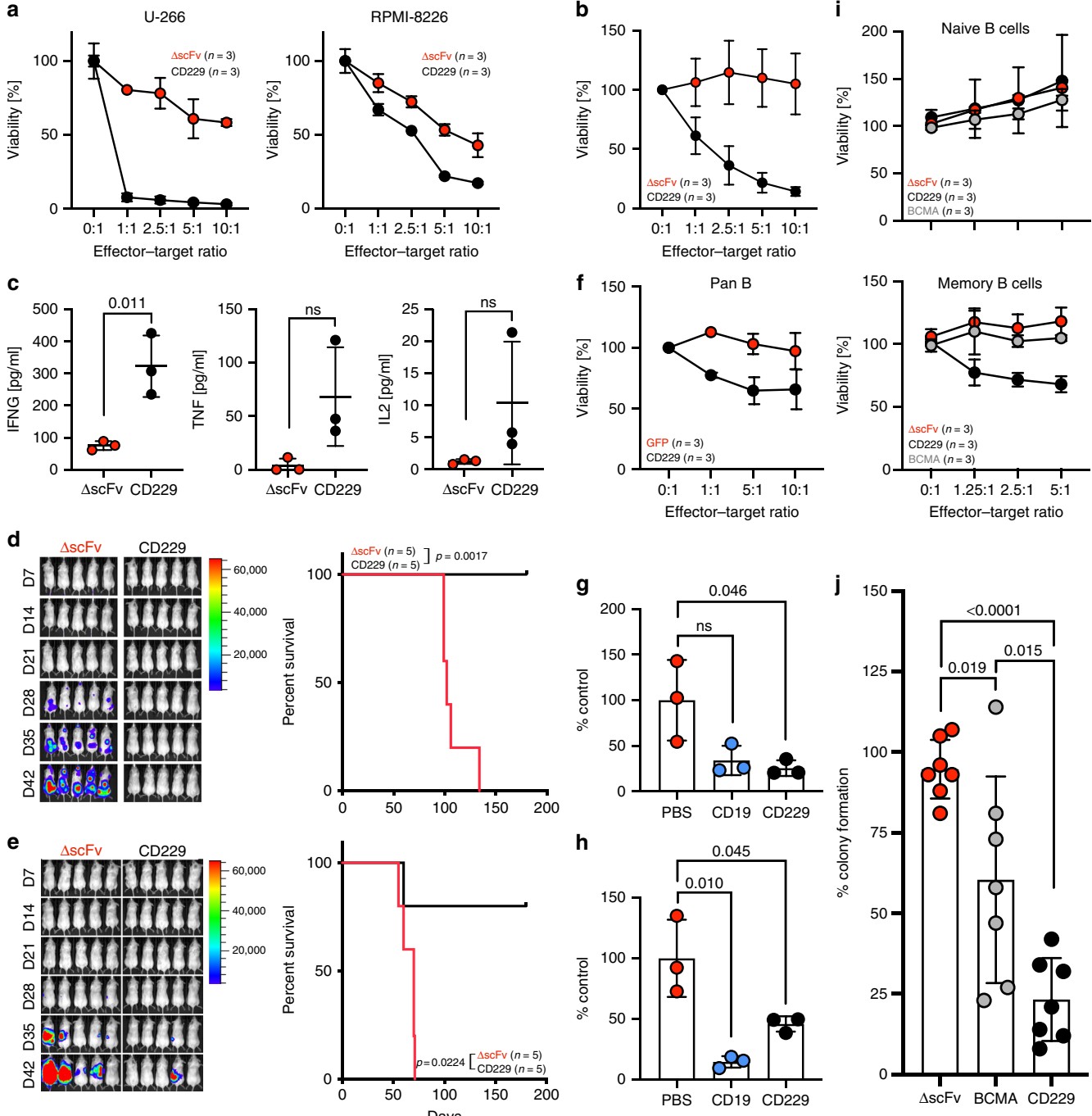

**Fig. 4 CD229 CAR T cells target MM plasma cells and MM-propagating cells. a** Killing of MM cell lines by CD229 CAR T cells or ΔscFv CAR T cells as determined by luciferase-based cytotoxicity assay after 16 h co-culture. Data represent mean ± SD from three independent experiments. **b** Cytotoxic activity of CD229 CAR T cells or ΔscFv CAR T cells against malignant cells from three patients with plasma cell leukemia as determined by flow cytometry after 2–4 h co-culture. Data indicate mean ± SD from three patient samples. **c** Secretion of cytokines by CAR T cells during co-cultures with primary plasma cell leukemia cells as determined by cytometric bead assay. Data indicate mean ± SD from three co-cultures. Significance was calculated using two-sided Student's t-test. NSG mice were injected with **d** $3 \times 10^6$ U-266 or **e** $1 \times 10^6$ RPMI-8226 cells expressing luciferase. After 1 week, mice were injected with $3 \times 10^6$ CAR T cells and bioluminescence was determined weekly. Differences in survival were determined by log-rank test. Data are representative of two independent experiments. **f** Killing of purified healthy B cells by CD229 CAR T cells or GFP T cells as determined by flow cytometry-based cytotoxicity assay. Data represent mean ± SD of three independent experiments. NSG mice were intravenously injected with $10^7$ healthy PBMCs. After two days, mice were injected with $1 \times 10^6$ CD19 or CD229 CAR T cells or PBS and sacrificed after 4 days. B-cell numbers were determined in the animals' **g** peripheral blood and **h** bone marrow. Data represent mean ± SD from three animals per group. Significance was determined by two-sided Student's t-test. **i** In vitro cytotoxicity of CD229, ΔscFv, or BCMA CAR T cells against purified memory or naive B cells as determined by flow cytometry. Data represent mean ± SD from three independent experiments. **j** Bone marrow mononuclear cells from seven MM patients were co-cultured with CD229, ΔscFv, or BCMA CAR T cells for 4–6 h, depleted of CD3+ CAR T cells, and plated in methylcellulose. Colonies were counted after 14–21 days of culture. Data represent mean ± SD from seven patient samples. Statistical significance was determined by two-sided Student's t-test. Source data are provided as a Source Data file.

functional CD229[neg/low] T cells. Furthermore, we showed that CD229 CAR T cells can efficiently be generated without measurable fratricide due to the downregulation of CD229 under the activating conditions present during manufacturing. However, it is possible that over time CAR T cells will upregulate CD229, which may limit their long-term persistence. The targeting of healthy CD229[high] T cells by CD229 CAR T cells needs to be addressed, e.g., by short-term expression of the CAR using RNA transfection instead of viral transduction, inclusion of a suicide gene to limit the persistence of CD229 CAR T cells, and combination with therapeutic strategies facilitating T-cell reconstitution, such as autologous stem cell transplantation. However, some residual hematopoietic toxicity may be acceptable in light of the comprehensive targeting of MM populations by this approach.

## Methods

**Patient samples and cell lines**. Peripheral blood, bone marrow, and unused CD34[+] hematopoietic stem cells from autologous transplant products were obtained from MM and/or plasma cell leukemia patients or healthy donors.

Cell lines U-266, RPMI-8226, 293T, and K562 cells were obtained from ATCC and cultured according to manufacturer's instructions. Competent bacteria TG1, BL21[DE3], and DH5α were obtained from Lucigen, Stbl3 cells were obtained from Thermo-Fisher.

**Flow cytometry analyses of CD229 expression**. Flow cytometry staining and analyses were performed as previously described[8]. To determine expression on MM plasma cells, bone marrow samples from patients with MM were stained with antibodies against CD38 and CD138 and co-stained with HLy9.1.25 (1:100), or 2D3 (50 μg/ml) and anti-FLAG (clone: M2, Sigma-Aldrich, 10 μg/ml). For some experiments, individual leukocyte populations were purified before staining (Stemcell Technologies). For a complete list of antibodies used for flow cytometry analyses see Supplementary Table 1. Commercially available antibodies were used at the dilutions recommended by the manufacturer. Gating schemas are provided as Supplementary Fig. 5.

**Antibody generation**. A human antibody phage display library was constructed as previously described[27] by cloning variable heavy and light chains from healthy donors separated by a $(Gly_4Ser)_3$ linker into a pUC119-based[28] phagemid containing the M13 leader and gene III. We performed two rounds of panning selections using recombinant human CD229 (R&D Systems, catalog number: 1898-CD-050) immobilized on MaxiSorp Immunotubes (Nunc). After confirming enrichment of CD229-specific binders in a polyclonal binding assay using time-resolved fluorescence (TRF), scFv sequences from the second selection round were cloned into pSANG10-3F[29] and transformed into BL21[DE3] cells (Lucigen). Monoclonal scFvs were expressed overnight in MagicMedia E. coli autoexpression medium (Thermo-Fisher) in 96-well plates and binding of individual supernatants to recombinant CD229 was determined by TRF. Plasmid DNA of binders was isolated using QIAprep Miniprep colums (Qiagen) and scFv sequences were determined by Sanger sequencing. For expression analyses, 2D3 was purified from 25 ml autoinduction cultures using NiNTA resin (Thermo-Fisher). For SPR analyses, scFvs were cloned into pBIOCAM5[27], scFv-Fc constructs expressed in 293F cells, and purified by NiNTA. For some experiments, 2D3 was expressed as a full IgG$_1$ antibody using Expi293 cells simultaneously transfected with individual pcDNA3.4 plasmids encoding light and heavy chains. Full IgG$_1$ antibodies were purified using Protein G (GE Healthcare) using standard protocols.

**Time-resolved fluorescence assay**. To determine binding of polyclonal and monoclonal antibodies, 5 μg/ml recombinant human CD229 was immobilized on black 96-well plates (Greiner Bio-One). Binding of antibodies was detected using anti-FLAG M2 (Sigma-Aldrich) followed by incubation with an anti-mouse IgG-Europium antibody (PerkinElmer). To determine relative binding of HLy9.1.25 and 2D3 to CD229, full IgG antibodies were immobilized and incubated with different concentrations of His-tagged recombinant CD229, which was detected by anti-His-Eu (PerkinElmer). After incubation with DELFIA Enhancement solution (PerkinElmer), TRF signal was determined on an EnVision plate reader (PerkinElmer).

**High-throughput surface plasmon resonance (SPR)**. A Xantec 200 m prism (CM5 analog) was removed from the freezer and brought to room termperature. For coupling, 100 μl of each of the 16 purified antibodies in scFv-Fc format was diluted to 20 μg/ml in 10 mM NaOAc pH 5.0 + 0.01% Tween-20. The continuous flow microspotter (CFM) was primed with 1x HBST (150 mM NaCl 10 mM HEPES + 0.01% Tween-20). The prism was first activated by cycling 12 mM sNHS, 3 mM EDC in 100 mM MES pH 5.0 for 5 min in the CFM. An anti-human Fc antibody (R&D Systems) was coupled for 7 min, followed by a 3 min rinse with

running buffer. The prism was immediately removed from the CFM and quenched in the MX96 imager with a 7 min injection of 0.5 M Ethanolamine. CD229-specific antibodies and soluble SLAM receptor proteins (R&D Systems) were diluted in phosphate-buffered saline (PBS) and injected sequentially at 200 nM to determine cross-reactivity. To determine binding constants recombinant human CD229 was injected at 200, 20, and 2 nM.

**Membrane proteome array specificity testing**. Integral Molecular, Inc. (Philadelphia, PA) performed specificity testing of 2D3 using the Membrane Proteome Array (MPA) platform. The MPA comprises 5,300 different human membrane protein clones (Supplementary Data 1), each overexpressed in live cells from expression plasmids that are individually transfected in separate wells of a 384-well plate[30]. The entire library of plasmids is arrayed in duplicate in a matrix format and transfected into HEK-293T cells, followed by incubation for 36 h to allow protein expression. Before specificity testing, optimal antibody concentrations for screening were determined by using cells expressing positive (membrane-tethered Protein A) and negative (mock-transfected) binding controls, followed by flow cytometric detection with an Alexa Fluor-conjugated secondary antibody (Jackson ImmunoResearch Laboratories). Based on the assay setup results, 2D3 (1.25 μg/ml) was added to the MPA. Binding across the protein library was measured on an Intellicyt HTFC (Ann Arbor, MI) using the same fluorescently labeled secondary antibody. To ensure data validity, each array plate contained positive (Fc-binding) and negative (empty vector) controls. Identified targets were confirmed in a second flow cytometric experiment by using serial dilutions of the test antibody. The identity of each target was also confirmed by sequencing.

**Manufacturing of CAR T cells**. We first replaced the PGK promoter in pRRLSIN.cPPT.PGK-GFP.WPRE with the human EF1A promoter. We then created a variant of this plasmid replacing GFP with an IGHV signal peptide, followed by the CD8α hinge and transmembrane domains, the 4-1BB costimulatory domain, and the CD3ζ signaling domain. Into this plasmid we cloned the different CD229-specific scFvs, the CD19-specific scFv FMC63[17], a BCMA-specific scFv[31], or a short 7aa linker fragment (ΔscFv) between the signal peptide and the hinge domain. Peripheral blood mononuclear cells (PBMC) from healthy donors were isolated by density-gradient centrifugation using Ficoll-Paque (GE). T cells were isolated from $10 \times 10^6$ PBMCs using CD3/CD28 T Activator beads (Thermo-Fisher) at a ratio of 1:3. PBMCs were incubated with beads for 2 h at room temperature after which T cells were isolated using magnetic separation and resuspended in AIM V medium (Thermo-Fisher)/5% human A/B serum (Sigma-Aldrich)/400IU IL-2 (R&D Systems). T cells were incubated for 48 h before transduction. On the day of the first transduction, half of the media was removed and 6 μg/ml polybrene was added to the cells together with concentrated lentiviral supernatants. After spinoculation for 1 h at $1000 \times g$, cells were resuspended and then returned to the incubator for another 24 h before repeating the transduction step. On the day after the transduction, 1 ml of complete growth media was replaced and IL-2 was renewed every 2 days. Cells were diluted to a concentration of $0.4 \times 10^6$/ml every 2 days. On day 12, T Activator beads were removed from the culture and cells were frozen in 90% fetal calf serum/10% dimethyl sulfoxide or used immediately for downstream assays. For some experiments, T cells underwent the production process without addition of lentiviral particles.

**Quantitative reverse transcription PCR (qRT-PCR)**. CD229 expression was measured by quantitative reverse transcription PCR. In brief, mRNA from sorted T cells and NK cells, as well as T cells that had undergone our CAR T-cell production process without addition of lentiviral particles was isolated using the RNeasy Mini kit (Qiagen), according to the manufacturer's instructions. Complementary DNA (cDNA) was generated using the SuperScript III First-Strand Synthesis system (Thermo-Fisher). We performed PCRs for CD229 (CD229_F: 5′-ATC ACC CCA ACC TCA CAT GC-3′; CD229_R: 5′-CGC ACA GAA GGC AAA CCA TC-3′) and GAPDH (GAPDH_F: 5′-ATT GCC CTC AAC GAC CAC TTT G-3′; GAPDH_R: 5′-TTG ATG GTA CAT GAC AAG GTG CGG-3′) using SYBR Green (Thermo-Fisher) and calculated the relative expression of CD229 to the samples' respective GAPDH expression.

**Western blot**. Total cell lysates of sorted lymphocyte populations and activated T cells were extracted using RIPA buffer (Thermo-Fisher). Protein concentrations were determined by BCA assay (Thermo-Fisher), samples separated by sodium dodecyl sulfate–polyacrylamide gel electrophoresis and separated proteins transferred to nitrocellulose membranes (GE Healthcare). Membranes were blocked with 5% non-fat milk-TBS and incubated with primary antibodies against CD229 (clone: HLy9.1.25) and ACTB (clone: BA3R, Thermo-Fisher). Membranes were washed and developed using a secondary anti-mouse IgG-HRP antibody (Jackson Immuno) and Amersham ECL Prime Western Blotting solution (GE Healthcare). Bands were visualized on a Gel Doc XR + System (Bio-Rad).

**T-cell proliferation assay**. In all, $2.5 \times 10^4$ CAR T cells were labeled with 1 nM CellTrace Far Red (Thermo-Fisher) according to the manufacturer's instructions and co-cultured with CD229-positive U-266 cells and CD229-negative K562 cells

in AIM V media for 72 h at 37 °C. Dye dilution as a measure of proliferation was determined using an LSRFortessa flow cytometer (BD).

**Flow cytometry-based cytotoxicity assay**. Two different flow cytometry-based cytotoxicity assays were used to determine cytotoxicity of CD229 CAR T cells against primary human cells. For normal lymphocytes and CD34[+] hematopoietic stem cells, target cells were purified (Stemcell Technologies) and stained with 20 μM calcein AM. Subsequently, $1 \times 10^4$ target cells were incubated with CD229 CAR T cells or control T cells at different effector-target ratios for 4 h at 37 °C. At the end of the co-culture, we stained the cells with propidium iodide, added counting beads (Life Technologies), and analyzed the samples on a FACSCanto or an LSRFortessa flow cytometer (BD). To determine cytotoxic activity of CD229 CAR T cells against plasma cell leukemia samples, $5 \times 10^4$ PBMCs containing >60% plasma cells were co-cultured with CAR T cells for 2 h or 4 h and then stained with antibodies against CD138, and CD56 or CD33 to identify malignant plasma cells, as well as 4′,6-diamidine-2′-phenylindole dihydrochloride (DAPI, Thermo-Fisher) and counting beads were added for normalization prior to analysis on an LSR Fortessa flow cytometer (BD). To determine B-cell subset cytotoxic activity of CD229 CAR T cells, $2.5 \times 10^4$ B cells were incubated with CAR T cells at different effector-target ratios for 4 h at 37 °C. We then stained with antibodies against CD19, CD27, and IgD to differentiate between memory and naive B cells, as well as DAPI. Counting beads were added for normalization prior to analysis on an LSRFortessa flow cytometer (BD).

**Luciferase-based cytotoxicity assay**. To determine cytotoxicity of CD229 CAR T cells against MM cell lines, U-266 and RPMI-8226 cells were transduced with pHIV-Luc-ZsGreen lentivirus and sorted on a FACSaria flow cytometer (BD) for GFP expression. Sorted MM cells were incubated with CD229 CAR T cells or control T cells at the indicated effector-target ratios for 4 h or overnight. At the end of the co-culture, cells were transferred to a black 96-well plate and incubated with 150 μg/ml D-luciferin (Gold Biotechnology Cat# LUCNA-2G) in PBS for 5 min before luminescence was determined on an EnVision plate reader (PerkinElmer).

**Multiple myeloma xenograft mouse models**. Six- to 8-week-old male NOD.Cg-Prkdc$^{scid}$ Il2rg$^{tm1Wjl}$/SzJ (NSG, The Jackson Laboratory (Cat#005557)) mice were sublethally irradiated with 200 cGy (Rad-Source RS-2000) and injected on the same day via the lateral tail vein with the indicated numbers of U-266 or RPMI-8226 cells stably expressing luciferase. On day 7 after tumor cell injection, the indicated numbers of CD229 CAR T cells, CAR T cells lacking a binding domain (StopX), or PBS alone were injected into the tail vein. Animals were weighed twice weekly and monitored for signs of distress in accordance with institutional regulations. For in vivo imagining, mice received an intraperitoneal injection of 3.3 mg D-luciferin. Photographic and luminescent images were acquired starting 10 min after the D-luciferin injection, both in prone and supine position using a PerkinElmer IVIS Spectrum. Myeloma progression was monitored every 7 days until the study endpoint. Average radiance (p/s/cm²/sr) was quantified for individual animals using Living Image software (PerkinElmer).

**In vivo cytotoxicity assay using human PBMCs**. Eight- to 11-week-old male NSG mice were intraperitoneally injected with two doses of 20 mg/kg busulfan (Selleck Chemicals, #S1692) on 2 consecutive days and on the following day injected with $10^7$ PBMCs isolated from healthy donors. On day 2 after PBMC injection, mice were injected with $10^6$ CD19 or CD229 CAR T cells or PBS via tail vein. On day 6 after PBMC injection, animals were sacrificed and spleen, bone marrow, and peripheral blood were collected for flow cytometry analysis. After a 5 min incubation in red blood cell lysis buffer (Biolegend), cells were washed twice in PBS, incubated with human and mouse FcR blocking reagents (Miltenyi Biotec) for 15 min, and then stained with population-specific antibodies (Supplementary Table 1) and DAPI for 30 min.

Stained samples were analyzed on an LSR Fortessa flow cytometer (BD). Lymphocyte numbers were normalized to the number of total human CD45[+] cells in each sample.

**Lambda light-chain enzyme-linked immunosorbent assay**. Serum was isolated from the peripheral blood from animals treated with CAR T cells at the time of sacrifice by centrifugation for 10 min at $400 \times g$. Lambda light-chain concentrations were determined using a commercial ELISA kit (Bethyl Laboratories).

**Cytometric bead assay**. Supernatants from primary co-culture cytotoxicity assays or CD3/CD28 bead-stimulated T cells were centrifuged for 10 min at $400 \times g$. Beads from a commercially available cytometric bead assay (BD) were incubated with supernatants and stained according to the manufacturer's instructions. Beads were analyzed on an LSRFortessa flow cytometer (BD) and results analyzed according to the manufacturer's instructions.

**Colony formation assay**. Bone marrow mononuclear cells were depleted of CD34[+] hematopoietic stem cells and co-cultured with or without ΔscFv, BCMA or CD229 CAR T cells for 4–6 h in RPMI 1640 containing 10% FBS. Cells were then magnetically depleted of CD3 + cells (Miltenyi Biotec) and then plated in methyl-cellulose[32]. MM colonies were quantified using an inverted microscope 14–21 days after plating.

**Study approval**. Samples were collected under tissue banking protocol # 45880, approved by the Institutional Review Board at the University of Utah. Written informed consent was obtained from participants prior to inclusion in the study.

All animal procedures were conducted under protocol #16-05007, approved by the Institutional Animal Care and Use Committee at the University of Utah.

**Statistics**. Significance of differences in survival were calculated by log-rank test. Significance of differences in cell numbers, cytokine levels, colony formation, and mean fluorescence intensity levels of CD229-specific antibodies were calculated by Student's $t$-test correcting for multiple comparisons using the Sidak-Bonferroni method were indicated. Significance of differences in lambda light-chain levels were calculated by Mann–Whitney test. All statistical tests were performed using Prism 7 (GraphPad Software). Results were considered significant when $p$ or adjusted $p < 0.05$.

## Data availability
The source data underlying Figs. 1A, 2A, 2E, 3A–E, 3G–L, and 4A–K are provided as a Source Data file. All relevant data are also available in the Article, Supplementary Information file of available from the corresponding author upon request.

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

## Acknowledgements

This project was supported by an International Myeloma Foundation Senior Grant (D.A.), an American Association for Cancer Research (AACR) fellowship (S.V.R.), a NCCN Young Investigator Award (T.L.), the Huntsman Cancer Institute Experimental Therapeutics program supported by the National Cancer Institute of the National Institutes of Health under Award number P30CA042014. (D.A. and T.L.), and by the DOD Era of Hope Scholar Expansion Award W81XWH-12-1-0077 (A.L.W.). We thank the Huntsman Cancer Institute in Salt Lake City, UT for the use of the Preclinical Research Resource (PRR), which performed the MM xenograft model, and the University of Utah Flow Cytometry core facility, which assisted with flow cytometry analyses and cell sorting. pUC119 was a gift from Joachim Messing (Addgene plasmid # 50010). pSANG10-3F (Addgene plasmid # 39264) and anti-Jagged_D5-pBIOCAM5 (Addgene plasmid # 39345) were gifts from John McCafferty. pRRLSIN.cPPT.PGK-GFP.WPRE was a gift from Didier Trono (Addgene plasmid # 12252). pHIV-Luc-ZsGreen was a gift from Bryan Welm (Addgene plasmid # 39196).

## Author contributions

S.V.R performed experiments, developed the luminescence-based in vitro cytotoxicity assay, analyzed data, and wrote the manuscript. T.L, conceived of the project, generated the antibody library, CD229-specific antibodies and CAR constructs, developed antibody screening assays, performed flow cytometry assays to determine CD229 expression on B lineage cells and correlative flow cytometry assays for the animal models, analyzed data, and wrote the manuscript. J.P. performed flow cytometry and analyzed data. P.D. performed flow cytometry assays determining 2D3 binding in patient bone marrow samples and analyzed data. E.R.V.M. and M.O. performed experiments and analyzed data. S.Y. established flow cytometry assays, helped establishing cytotoxicity assays, and analyzed data. S.D.S. established and performed the humanized mouse model, performed flow cytometry analyses, and analyzed data. Y.A. established experimental design and performed high-throughput surface plasmon resonance assays, and analyzed data. K.D.L. and R.R.M. established and validated the flow cytometry assays determining 2D3 binding in patient bone marrow samples and analyzed data. W.M. performed colony formation assays and analyzed data. A.W. developed the humanized mouse model and analyzed data. D.A. conceived of the project, analyzed data, and wrote the manuscript.

## Competing interests

D.A., T.L. and S.V.R. are inventors on PCT application US2017/42840 "Antibodies and CAR T Cells for the Treatment of Multiple Myeloma" describing the therapeutic use of CD229 CAR T cells for the treatment of multiple myeloma. The other authors declare no competing interests.
