## [Peer Review File · Nature Communications]

Reviewers' comments:

Reviewer #1 (Remarks to the Author): Expert in myeloma treatment

Authors report pre-clinical development of a CAR against CD229, a target previously investigated by the authors in MM. Authors describe the creation of a fully human scFv and describe activity against MM cells and other hematopoietic cells. As authors highlight, there continues to be unmet clinical need in MM; even the most advanced MM therapies do not seem to be curative, including (unfortunately) anti-BCMA CAR T cells. Authors propose that anti-CD229 CAR T cells could improve upon anti-BCMA results by co-targeting clonotypic B cells, and their results provide preliminary support for this, showing that their CAR targets certain B cell subsets and more stringently depletes MM colony formation compared to anti-BCMA CARs. The liabilities of the approach are targeting of T cells and sparing of target-dim cells. Authors provide some reassuring data that residual CD229-low non-CAR T cells retain core functions (by in vitro assays) and that there is not fratricide during CAR T cell mfg, probably due to down-regulation of CD229 on activated T cells. These data expose, however, the resistance of target-dim cells to their CAR, which is also seen in their MM cell line data, where the CD229-low RPMI-8226 line appears less sensitive than the CD229-high U-266 line in vitro. And fratricide might yet be a problem in vivo for late-persisting CAR T cells, which one would like to survive after initial in vivo expansion to provide long-term surveillance. At this point, the significance of these liabilities needs to be tested clinically, and I don't think these liabilities would prevent entry of this CAR to clinical trials in appropriately selected and consented patients, nor do I think further pre-clinical modeling of these toxicities at this stage would be worthwhile. Indeed, there are currently CAR T cells in clinical trials for myeloma patients targeting CD38, which has far greater liabilities with respect to presence on normal T cells and other hematopoietic cells, including myeloid progenitors. I would like to see the above liabilities discussed more forthrightly in the discussion, however, particularly the problem of target-dim MM cells being relatively resistant to their CAR.

Minor points:

- Intro mentions 30,000 new MM cases per year. I believe this is the # of new cases in the US alone, so this should be clarified.
- Intro mentions that "BCMA-negative relapses occur often" after anti-BCMA CARs. This is unsupported by the cited studies. I don't think Raje et al reported on BCMA expression on MRD or at relapse. Cohen et al showed diminishment of BCMA expression on residual disease after initial CAR expansion, but BCMA expression later rebounded in most cases, at least by flow cytometry. It might be more accurate to say that modulation of BCMA has been observed after anti-BCMA CAR therapy, but significance is still uncertain. Authors are right to point out, however, that BCMA expression is heterogeneous, to the point that many patients were excluded from trials due to low/absent BCMA expression.
- Authors should state the source of the anti-BCMA scFv used in their comparative experiments.
- Last sentence in discussion: I suggest avoiding mention of "curative potential" in a preclinical paper, given the humbling clinical experience with every therapeutic advance in MM over the last many decades.

Reviewer #2 (Remarks to the Author): Expert in CAR-T cells

In the submitted the authors describe the generation of CD229-CAR T cells for the cell therapy of multiple myeloma (MM). They explore the effector function of CD229-CAR T cells in vitro and in vivo and demonstrate potent anti MM activity. The manuscript is well written, and the authors' conclusion are supported by the presented data. However, I have several major and minor concerns the authors should address.

Major concerns:

1. Animal experiments: it seems that each animal experiment was only performed once; please provide data for additional animals to substantiate the promising results shown in Figure 4.
2. CAR expression: I might have missed it, but I could not find that the authors demonstrate CAR expression of transduced T cells – please provide data for all CAR constructs used in this study.
3. CEFT Elispot assay: the experimental set up is unclear, important controls are missing and the results in its present form are impossible to interpret; recommend removing from manuscript.

Minor concerns:

1. Humanized mouse model: calling a model in which human PBMCs are injected into NSG mice after 2 doses of busulfan, injecting CAR T cells on day 2 and harvesting cells on day 6, a humanized mouse model is misleading unless the authors can show engraftment of human CD34+ progenitor cells. With no engraftment, recommend calling this experiment an 'in vivo cytotoxicity assay with human PBMCs'.
2. Figure S1C: believe that 41BB should be before the CD3z signaling domain in the scheme of the CAR.

Reviewer 1

Comment 1.1: "I would like to see the above liabilities discussed more forthrightly in the discussion, however, particularly the problem of target-dim MM cells being relatively resistant to their CAR"

Reply: We have added a paragraph discussing the possibility of antigen escape as well as a sentence clarifying the possibility of the upregulation of CD229 on CD229 CAR T cells over time potentially affecting their long-term persistence.

Comment 1.2: "Intro mentions 30,000 new MM cases per year. I believe this is the # of new cases in the US alone, so this should be clarified."

Reply: We have changed this sentence in the introduction as requested.

Comment 1.3: "Intro mentions that "BCMA-negative relapses occur often" after anti-BCMA CARs. This is unsupported by the cited studies. I don't think Raje et al reported on BCMA expression on MRD or at relapse. Cohen et al showed diminishment of BCMA expression on residual disease after initial CAR expansion, but BMCA expression later rebounded in most cases, at least by flow cytometry. It might be more accurate to say that modulation of BCMA has been observed after anti-BCMA CAR therapy, but significance is still uncertain. Authors are right to point out, however, that BCMA expression is heterogeneous, to the point that many patients were excluded from trials due to low/absent BCMA expression."

Reply: We have changed this sentence in the introduction accordingly.

Comment 1.4: "Authors should state the source of the anti-BCMA scFv used in their comparative experiments."

Reply: We included references describing the sources of the CD19 and BCMA-specific scFvs in the method section under "Manufacturing of CAR T cells".

Comment 1.5: "Last sentence in discussion: I suggest avoiding mention of "curative potential" in a preclinical paper, given the humbling clinical experience with every therapeutic advance in MM over the last many decades."

Reply: We have changed this sentence in the discussion as requested.

Reviewer 2

Comment 2.1: “Animal experiments: it seems that each animal experiment was only performed once; please provide data for additional animals to substantiate the promising results shown in Figure 4.

Reply: We have now repeated both xenograft experiments as described in our manuscript and observed comparable efficacy of CD229 CAR T cells against multiple myeloma cell lines U266 (Fig. 1A) and RPMI8226 (Fig. 1B) by *in vivo* imaging. For the more rapidly growing RPMI8226 cell line, we also provide survival data (Fig. 1C). If requested, we can also provide survival data for the U266 experiment, which, however, will require a much longer follow-up period.

Comment 2.2: “CAR expression: I might have missed it, but I could not find that the authors demonstrate CAR expression of transduced T cells – please provide data for all CAR constructs used in this study.”

Reply: We have now added CAR expression data as Suppl. Fig. 2C.

Comment 2.3: “CEFT Elispot assay: the experimental set up is unclear, important controls are missing and the results in its present form are impossible to interpret; recommend removing from manuscript.”

Reply: We have removed these data from the manuscript. Alternatively, if requested, we could also include additional controls or provide further clarification.

Comment 2.4: “1. Humanized mouse model: calling a model in which human PBMCs are injected into NSG mice after 2 doses of busulfan, injecting CAR T cells on day 2 and harvesting cells on day 6, a humanized mouse model is misleading unless the authors can show engraftment of human CD34+ progenitor cells. With no engraftment, recommend calling this experiment an ‘in vivo cytotoxicity assay with human PBMCs’.

Reply: We have changed the description of this experiment throughout the manuscript as requested.

Figure 1: *In vivo* efficacy of CD229 CAR T cells against multiple myeloma. NOD.Cg-Prkdc^{scid} Il2rg^{tm1Wjl}/SzJ mice were irradiated with 200cGy and injected with (A) 3×10^6 U266 or (B) 1×10^6 RPMI8226 expressing luciferase. After 1 week, mice were injected with 3×10^6 CAR T cells and bioluminescence was determined weekly using *in vivo* imaging. Statistical significance of differences in survival determined for mice engrafted with RPMI8226 cells (C) were determined by log-rank test.

Comment 2.5: “Figure S1C: believe that 41BB should be before the CD3z signaling domain in the scheme of the CAR.”

Reply: Thank you for pointing out this mistake, we have corrected the figure accordingly.

Again, we would like to thank both reviewers for their helpful and constructive comments and hope that our revised manuscript is now acceptable for publication in *Nature Communications*.

REVIEWERS' COMMENTS:

Reviewer #1 (Remarks to the Author):

All my comments have been adequately addressed.

Reviewer #2 (Remarks to the Author):

The authors have addressed all my concerns appropriately.